# The Relationship between Time, Race, and Estrogen Receptor Alpha in Estradiol-Induced Dermal Fibrosis

**DOI:** 10.3390/biomedicines12010182

**Published:** 2024-01-15

**Authors:** DeAnna Baker Frost, Alisa Savchenko, Carol Feghali-Bostwick, Bethany Wolf

**Affiliations:** 1Department of Medicine, Division of Rheumatology and Immunology, Medical University of South Carolina, 96 Jonathan Lucas Street, Charleston, SC 29425, USA; bakerde@musc.edu (D.B.F.); feghalib@musc.edu (C.F.-B.); 2Chobanian & Avedisian School of Medicine, Boston University, 72 E. Concord Street, Boston, MA 02118, USA; savchenko.alisa96@gmail.com; 3Department of Public Health Sciences, Medical University of South Carolina, 135 Cannon Street, Room 305F, Charleston, SC 29425, USA

**Keywords:** estradiol, estrogen receptor alpha, fibrosis, skin

## Abstract

In the skin, estradiol (E2) promotes profibrotic and proinflammatory cytokines, contributing to extracellular matrix (ECM) deposition. However, the magnitude of the response differs. Using the human skin organ culture model, we evaluated donor characteristics and correlations that contribute to E2-induced *interleukin-6* (*IL-6*), *transforming growth factor beta 1* and *2* (*TGFB1* and *TGFB2*), *collagen IA2* (*Col IA2*), *collagen IIIA1* (*Col IIIA1*), and *fibronectin* (*FN*) expressions. In vehicle- and E2-treated dermal skin tissue transcripts, we confirm differences in the magnitude; however, there were positive correlations between profibrotic mediators and ECM components 48 h after E2 treatment. Also, positive correlations exist between baseline and E2-induced *TGFB1*, *IL-6*, *Col IIIA1*, and *FN* transcripts. Since estrogen receptor alpha (ERA) can propagate E2′s signal, we measured and detected differences in its baseline and fold change transcript levels, with a significant decline in baseline levels 48 h after incubation and an increase 48 h after E2 treatment. There was a trend to higher transcript levels in African American donors 24 h earlier. Finally, E2-induced *ERA* transcript levels negatively correlated with its own baseline levels and positively correlated with *FN*, *TGFB1*, and *Col IA2* transcript levels. Therefore, our data suggest ERA, E2 exposure time, and race/ethnicity contribute to E2-induced dermal fibrosis.

## 1. Introduction

Estrogen is one of the hormones responsible for secondary sex characteristics, but it can also affect dermal thickness. Estradiol (E2), one of the major circulating forms of estrogen [1], promotes extracellular matrix (ECM) production [2,3,4], contributing to skin elasticity and diminished wrinkling [5]. E2 also stimulates transforming growth factor beta 1 and 2 (TGFB1 and TGFB2) production [4,6], which are known profibrotic cytokines, and the proinflammatory cytokine interleukin-6 (IL-6) [7]. TGFB1, TGFB2, and IL-6 induce ECM deposition [8,9] and contribute to fibrosis. Therefore, through proinflammatory and profibrotic cytokines and ECM production, E2 participates in inflammation and dermal fibrosis.

Several models are used to study dermal fibrosis. One such model, the human skin organ culture model, is advantageous and favored over human dermal fibroblasts in vitro since the dermal, epidermal, and ECM layers remain intact, allowing for cell–cell interactions in their natural structure [10]. While the human skin organ culture model has proven invaluable in translational research, variability exists in the magnitude of response to stimulants [11,12]. Recent studies found the anatomic location, baseline gene transcript levels, and age of the donor impact the responsiveness of dermal tissue to TGFB1 [13], defining a relationship between donor characteristics and phenotypic response.

Studies suggest that donor age may also influence the dermal response to E2. As systemic estrogen levels decline in post-menopausal women, estrogen production shifts to extra-gonadal organs, leading to dermal wrinkling, thinning, and laxity due to decreased collagen production [14]. But, these clinical observations improve with exogenous E2 supplementation [15].

The donor’s self-identified race/ethnicity may also affect dermal responsiveness to E2. African American (AA) post-menopausal women have decreased skin wrinkling when compared to non-AA women [16] and continued to have significantly less skin wrinkling after receiving E2-based hormone therapy [17], likely the result of enhanced ECM production in the skin. Yet, it is unclear if any donor characteristics directly influence E2-treated skin tissue.

In addition to donor characteristics, E2 receptor signaling may modify dermal ECM production in response to E2 treatment. Estrogen receptor alpha (ERA) is one of the major receptors responsible for cellular E2 signal propagation [18]. ERA signaling likely occurs in dermal tissue since human dermal fibroblasts express ERA both in the cytoplasm and nucleus [19]. A relationship in human skin exists between E2 signaling through ERA and ECM production [3,4]. In mouse skin, data suggest that estrogen receptors regulate collagen biosynthesis [20]. While ERA modulates ECM components in breast cancer [21] and its expression is used to guide breast cancer treatment [22], there is no established association between *ERA* and ECM steady-state transcript levels in E2 treated-dermal tissue. 

In this study, we examined associations between donor characteristics and the E2-induced proinflammatory and profibrotic mediator’s steady-state transcript levels using the human skin organ culture model. Additionally, we evaluated correlations between *ERA* and ECM steady-state transcript levels in dermal tissue. These data will provide insight into factors which contribute to dermal responsiveness to E2 stimulation. 

## 2. Materials and Methods

### 2.1. Ex Vivo Human Skin Organ Culture Model

We received 34 fresh, skin tissue samples from discarded skin after completing skin-resection procedures. The donors’ demographic information is summarized in Table 1. The ex vivo human skin organ culture model was used as previously described [3,11,23,24]. For experimentation, we used six-well tissue culture dishes (Costar, Corning, NY, USA) with six 3 mm punches/well placed dermal side down in low estrogen conditions (serum-free, phenol-red-free Dulbecco’s Modified Eagle Medium (DMEM), Cytiva, Marlborough, MA, USA). Each skin sample was treated with E2 (Tocris, Minneapolis, MN, USA) at 10 nM or with an equal volume of 100% ethanol (ETOH, Hyclone, South Logan, UT, USA), serving as the vehicle for E2. At the appropriate time, the dermal tissue was harvested and stored at −80 °C until use. 

### 2.2. Measurement of Steady-State Transcript Levels Using Quantitative PCR

Total RNA was isolated from human skin tissue through homogenization using TRIzol (Invitrogen, Carlsbad, CA, USA), with chloroform extraction, precipitation using 100% isopropanol, washing in 75% ethanol, and resuspension in RNAse-free water. Steady-state transcript levels were measured using quantitative PCR (qPCR) and levels were normalized to *beta-2 microglobulin* (*B2M*, Hs00187842_m1) or *GAPDH* (Hs02758991_g1). Primers specific for *fibronectin* (*FN*, Hs00365052_m1), *TGFB1* (Hs00998133_m1), *TGFB2* (Hs00234244_m1), *collagen IIIA1* (*Col IIIA1*, Hs00943809_m1), *collagen IA2* (*Col IA2*, Hs00164099_m1), *estrogen receptor alpha* (*ERA*, Hs_0017486_m1) and *IL-6* (Hs00985639_m1) were all purchased from Thermofisher Scientific (Rockford, IL, USA). The delta-delta-CT method was used to determine the fold change in the gene of interest in E2-treated dermal tissue as compared to vehicle treatment. The delta-CT method was used to determine the relative quantity representing the baseline steady-state levels in vehicle-treated tissue. 

### 2.3. Statistical Analysis

Descriptive statistics were calculated for the donors’ characteristics. Differences in gene expression over time reported as relative quantity and fold change relative to baseline, were evaluated using a linear mixed model (LMM) approach. Models included a fixed effect for time and a random subject effect to account for correlation between measures of gene expression from the same subject collected over time. Model assumptions were checked graphically, and a natural log transformation was used to address issues of non-normality. Differences in gene expression between time points were evaluated using linear contrasts from the model. Differences in gene expression by race within timepoint were also examined in additional LMMs that included fixed effects for race, time, and the race-by-time interaction. Notably, only donors of Caucasian and AA ethnicities were considered in these models due to the paucity of samples for other groups. The correlation between expression levels for the different genes measured was also examined using the generalized R^2^ approach proposed by Nakagawa and Schielzeth to estimate an overall measure of correlation accounting for the relationship over time [25]. *p*-values for all associations between genes were Bonferroni adjusted for pairwise comparisons. Statistical significance was determined as a *p*-value < 0.05. One donor was missing age which was imputed using the median age of the study population. Additionally, the study population included only one male. Therefore, sensitivity analyses were conducted considering the results of all statistical evaluations excluding the donor with missing age and the male.

## 3. Results

### 3.1. Demographics

Table 1 includes the donor’s demographic information for the dermal tissue used for experimentation. All donors were >18 years old, with the majority of donors between 30 and 50 years old. The largest segment of donors self-identified as Caucasian (67.6%), with the remaining self-identified as AA (26.5%), Hispanic (2.9%), and Cambodian (2.9%). The percentage of donors that identified as males or females was 2.9% and 97.1%, respectively. 

### 3.2. Variability in Steady-State Proinflammatory and Profibrotic Transcript Levels in Vehicle- and Estradiol-Treated Dermal Tissue

While the human skin organ culture model is considered the gold standard for skin research, this model has high intra-and inter-sample variability [11,12]. Through measuring the proinflammatory and profibrotic components induced by E2 treatment [2,3,4], we examined differences in the quantity of profibrotic and proinflammatory steady-state transcript levels in non-E2 treated samples. Using the proinflammatory and profibrotic components that were previously measured in dermal tissue samples [4], we measured each gene at the 24-, 48-, and 72 h time points and found gene-dependent differences in transcript levels prior to E2 treatment (Table 2). Even though *TGFB1, Col IIIA1,* and *IL-6* vehicle-treated steady-state transcript levels were the most abundant (Table 2), there was not a statistically significant difference in transcript levels of each gene over time. Similarly, we detected differences in the calculated fold change in each gene in response to E2 treatment, with the majority of the fold changes above baseline occurring after 48 h of treatment (Table 3). Therefore, the number of steady-state transcript levels identified differed depending on the gene measured and the timing of E2 stimulation.

### 3.3. Variability in Steady-State ERA Transcript Levels in Vehicle and Estradiol-Treated Dermal Tissue

Because we detected differences in the vehicle-treated steady-state transcript levels of proinflammatory and profibrotic mediators, we assessed steady-state transcript levels of *ERA* since E2 receptor signaling is involved in E2-induced ECM components in human skin [3,4]. As in the other measured genes, we also found differences in the number of vehicle-treated transcript levels of *ERA* (Table 2). Interestingly, after 48 h, there was a significant decrease in mean *ERA* steady-state transcript levels in vehicle-treated dermal tissue compared to 24 h. However, after 72 h, there was a trend toward an increase in the mean steady-state transcript levels, though statistical significance was not achieved (Figure 1a). We also observed differences in E2-induced *ERA* steady-state transcript levels (Table 3), with a significant increase in the mean transcript levels 48 h after E2 treatment and a trend to a significant decrease after 72 h (Figure 1b). A sensitivity analysis excluding the male donor yielded similar results for the relative quantity of *ERA* over time. However, fold change in *ERA* between 48 and 72 h became significant at *p* < 0.05.

### 3.4. Correlation between ERA Steady-State Transcript Levels and Race/Ethnicity and Donors’ Age in Estradiol-Treated Dermal Tissue

Next, we evaluated if these transcript levels varied according to the donor’s characteristics, namely the race/ethnicity and age of the donor. While there were no significant correlations between the self-identified race and the steady-state transcript levels of the proinflammatory and profibrotic genes at any time point, self-identified AA donors trended to produce more E2-induced *ERA* steady-state transcript levels 24 h after E2 treatment compared to self-identified Caucasian donors (Figure 2), which is 24 h earlier than overall donors (Figure 1b). The results were similar to a sensitivity analysis excluding the male donor (*p* = 0.101). We also investigated the effect of age on E2-induced profibrotic gene variation. Although studies document that ECM proteins evolve in humans over different ages [26], we did not find any significant associations between baseline or E2-induced steady-state transcript levels of ECM components and the age of the donors, and this remained non-significant when excluding the one participant missing age.

### 3.5. Correlations between Estradiol-Induced Profibrotic Mediators and Baseline ERA Steady-State Transcript Levels

Due to the gene-dependent differences in the dermal tissue response to E2 (Table 3), we questioned if this is a result of baseline *ERA* expression in the skin since E2 can use ERA for signal transduction. Thus, we examined the relationship, if any, between baseline *ERA* and E2-induced profibrotic mediator transcript levels. There was a significant negative correlation between baseline and E2-induced *ERA* steady-state transcript levels after 48 h (r = −0.457), but no significant correlations between baseline *ERA* and E2-induced profibrotic mediator transcript levels (Table 4). Although the results were similar in a sensitivity analysis excluding the male donor, the association between *TGFB1* and *TGFB2* became significant (adj *p* = 0.039), and the association between *TGFB1* and *ERA* was no longer significant (*p* = 0.076).

### 3.6. Correlations between E2-Induced Profibrotic Mediators and E2-Induced ERA Steady-State Transcript Levels

Figure 1b and Figure 2 demonstrate that E2-induced *ERA* steady-state transcript levels differ based on E2 exposure time and the self-identified race/ethnicity of the donor. Similarly, we described that the baseline *ERA* transcript levels were negatively correlated with E2-induced *ERA* levels (Table 4). Therefore, we examined if E2-induced *ERA* steady-state transcript levels correlate with E2-induced profibrotic mediator steady-state transcript levels in dermal tissue. In Table 4, we found significant positive correlations between E2-induced *ERA* and *FN* (r = 0.347), *TGFB1* (r = 0.315), and *Col IA2* (r = 0.382) transcript levels.

### 3.7. Correlations between Estradiol-Induced Profibrotic Mediator Steady-State Transcript Levels

Since the preponderance of the genes with a fold change above baseline occurred 48 h after E2 treatment (Table 3), we further examined for correlations between profibrotic steady-state transcript levels 48 h after E2 treatment. Even though *IL-6*, *TGFB1*, *TGFB2*, *Col IA2*, *Col IIIA1*, and *FN* are individually and variably induced by E2 [4,7], we discovered several correlations between these E2-induced profibrotic mediators. There were significant positive correlations between *FN* and *Col IA2* (r = 0.678), *TGFB1* (r = 0.550), and *Col IIIA1* (r = 0.499) steady-state transcript levels 48 h post E2 treatment (Table 4). We also found significant correlations between *TGFB1* and *Col IA2* (r = 0.471) and between *Col IA2* and *Col IIIA1* (r = 0.439) steady-state transcript levels (Table 4). 

### 3.8. Correlations between Vehicle-Treated and Estradiol-Treated Profibrotic Mediator Steady-State Transcript Levels

A previous study demonstrated a significant negative correlation between baseline and TGFB1-induced *FN* steady-state transcript levels [13]. Therefore, we examined for correlations between the vehicle-treated and E2-treated steady-state transcript levels. We found moderate but significant positive correlations between the vehicle-treated and E2-treated steady-state levels of *FN* (r = 0.219), *TGFB1* (r = 0.279), *IL-6* (r = 0.250), and *Col IIIA1* (r = 0.208) (Table 5). The results were similar in sensitivity analyses excluding the male donor.

## 4. Discussion

Several studies confirm that dermal tissue responds to E2 by increasing profibrotic mediators at both the transcript and protein levels [2,3,4,6], establishing the relationship between E2 and fibrosis. Yet, this is the first study to investigate the influence of donor characteristics and *ERA* steady-state transcript levels on E2-induced proinflammatory and profibrotic mediators. For experimentation, we chose the human skin organ culture model since it is the gold standard, and we have successfully used this model to evaluate the effects of profibrotic and proinflammatory stimulants on human skin tissue [3,4,11,23,24,27,28,29]. 

We confirmed that not all dermal tissue produced steady-state transcript levels at the same magnitude before or after E2 treatment, suggesting additional factors likely affect steady-state transcript levels. Other studies describe differences in the magnitude of profibrotic mediators depending on the stimulants used [30]. Therefore, in this manuscript, we investigated correlations between the steady-state transcript levels of the profibrotic and proinflammatory genes analyzed. 

We found significantly lower baseline *ERA* levels after 48 h in culture, suggesting that *ERA* steady-state transcript levels are dynamic and negatively influenced by time. In cultured endothelial cells, baseline ERA protein levels change over time [31]. Interestingly, the *ERA* expression in adipose tissue also decreases with the age of the donor [32], although we did not find correlations between *ERA* transcript levels and the age of the donor in dermal tissue. Despite the significant decrease in mean *ERA* steady-state transcript levels at 48 h, those levels significantly increased with E2 treatment. While it has been reported that E2 increases ERA expression in the posterior vaginal wall [33], endothelial cells [31], and dermal fibroblasts [3], we are the first to report a relationship between E2-induced *ERA* transcripts and low baseline *ERA* transcripts. A similar response is observed in dermal tissue with low baseline steady-state transcript levels of *FN*, which produce significantly higher levels after TGFB1 stimulation [13]. 

Even though we did not find that age was correlated with E2-induced differential expression in these genes, we found that the donors’ self-identified race alters *ERA* transcript levels. AA donors have a trend to earlier increase in *ERA* transcripts by E2. Ethnic differences in mRNA *ERA* levels are described in gluteal adipose tissue, with higher levels identified in AA subjects [34]. Previous studies also found in the skin that differential expression of several genes which are implicated in fibrosis was influenced by the AA race and weight of the donor [35]. But this is the first study to examine ethnic differences in *ERA* steady-state transcript levels in dermal tissue. Several studies found that *ERA* gene alterations and polymorphisms are associated with type II diabetes, metabolic syndrome, and uterine leiomyomas in AA patients [36,37,38]. Future studies include examining if polymorphisms exist in dermal tissue, leading to higher *ERA* transcript levels.

There was also a negative correlation between baseline and E2-induced *ERA* steady-state transcript levels, which confirms the results in Figure 1. These data suggest that E2 stimulation increases the *ERA* transcript abundance and thereby its accessibility in the skin. More positive correlations were found between E2-induced *ERA* and profibrotic steady-state transcript levels compared to baseline *ERA*, i.e., between *TGFB1*, *FN*, and *Col IA2* steady-state transcript levels. Therefore, E2 may prime skin tissue to produce more profibrotic steady-state transcript levels that require ERA as the receptor to propagate E2′s signal. Animal models suggest the importance of ERA expression in the skin during the wound-healing process, which includes ECM regulation and production [39]. Wild-type male mice treated with E2 developed a significant increase in their epidermal thickness, compared to mice with functionally absent ERA [40]. Additionally, male mice treated with fulvestrant, an ERA inhibitor, displayed lower collagen content and a lower pulmonary fibrosis score in a bleomycin-induced pulmonary fibrosis model [41]. Finally, in humans, dermal fibroblasts and tissue treated with E2 and fulvestrant displayed significant decreases in fibrotic transcript and protein levels compared to treatment with E2 and vehicle [3,4]. Therefore, it is established in the literature that ERA contributes to ECM production to promote fibrosis. Our data suggest a similar positive relationship between *ERA* transcript levels and ECM in human skin because of E2 stimulation. 

In vehicle-treated dermal tissues, we identified gene-specific differences in the steady-state transcript levels and modest but significant gene-specific positive correlations between their vehicle-treated and their E2-induced steady-state transcript levels. Taken together, we identified gene-specific correlations between their E2-induced ECM transcript levels and two variables: 1. their baseline ECM transcript levels and/or 2. E2-induced *ERA* transcript levels. While E2-induced *IL-6* and *Col IIIA1* transcript levels only positively correlate with their baseline levels, E2-induced *Col IA2* transcript levels only correlate with E2-induced *ERA* transcript levels. However, E2-induced *FN* and *TGFB1* transcript levels positively correlate with both their baseline and E2-induced *ERA* transcript levels. Yet, *TGFB2* transcript levels do not correlate with either its baseline or E2-induced *ERA* transcript levels. These data imply that some ECM transcripts may be contingent on their baseline transcript level, utilize E2-induced *ERA* transcripts, and may utilize both or neither. As a result, each scenario influences E2-induced profibrotic mediator transcripts in dermal tissue. 

Despite the gene-specific differences in the amount of steady-state transcript levels after E2 stimulation, we also identified positive correlations between profibrotic genes. In idiopathic pulmonary fibrosis, lung fibroblasts that are exposed to ECM and isolated from diseased lung tissue can activate new ECM gene translation [42]. Therefore, there is a precedence in the lung that formed ECM can perpetuate fibrosis. While we reported that E2-induced profibrotic genes individually promote fibrosis in human dermal tissue [3,4], our current data imply that the positive correlations between the steady-state transcript levels of these E2-stimulated profibrotic genes may together contribute to dermal fibrosis. 

The limitations of this study include the inability to determine the contribution of menopausal status and hormonal supplementation in the donors, as both can affect dermal thickness through the ECM [43]. Overall, the small sample size may have affected the strength of the results, and we were not able to assess the effects of sex on E2-induced proinflammatory and profibrotic mediators or *ERA* steady-state transcripts in dermal tissue due to the paucity of male donor samples. While the focus of this study was on steady-state transcript levels, we did not evaluate protein levels, although E2 does increase the protein levels of profibrotic mediators [2,3,4,7]. The race/ethnicity of the donors were self-identified. Finally, we did not assess the influence of other sex hormones on steady-state proinflammatory or profibrotic steady-state transcript levels.

In conclusion, despite the differences noted, we found positive correlations between profibrotic mediators and ECM components. E2 significantly increases *ERA* steady-state transcript levels in the skin. However, the differences in the timing of E2-induced *ERA* levels in AA donors may contribute to variations in dermal tissue response to E2. E2-induced *ERA* transcripts result in positive correlations with *TGFB1*, *FN*, and *Col IA2* steady-state transcript levels. Finally, ECM transcripts positively correlate with each other. Taken together, multiple elements positively impact E2-induced dermal fibrosis. 

## Figures and Tables

**Figure 1 biomedicines-12-00182-f001:**
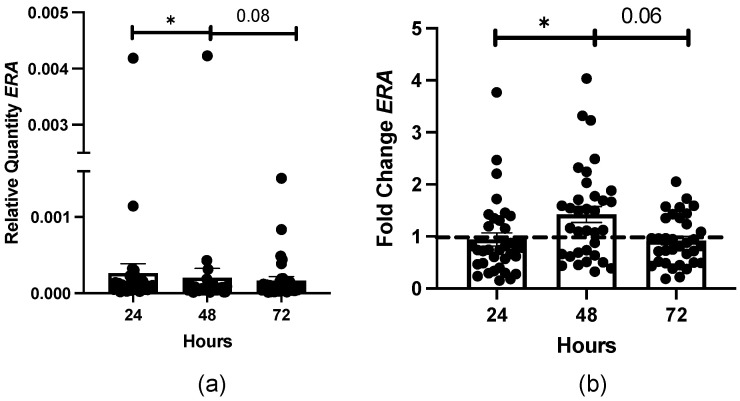
Baseline and E2-induced *ERA* transcript levels in dermal tissue over time. (**a**) Mean relative quantity of baseline *ERA* transcript levels at 24, 48, and 72 h. (**b**) Mean fold change in E2-induced *ERA* steady-state transcript levels at 24, 48, and 72 h. Dotted line represents vehicle-treated group normalized to 1. Statistical testing was performed using a linear mixed model (LMM) approach, with linear contrasts from the model. * *p* < 0.05.

**Figure 2 biomedicines-12-00182-f002:**
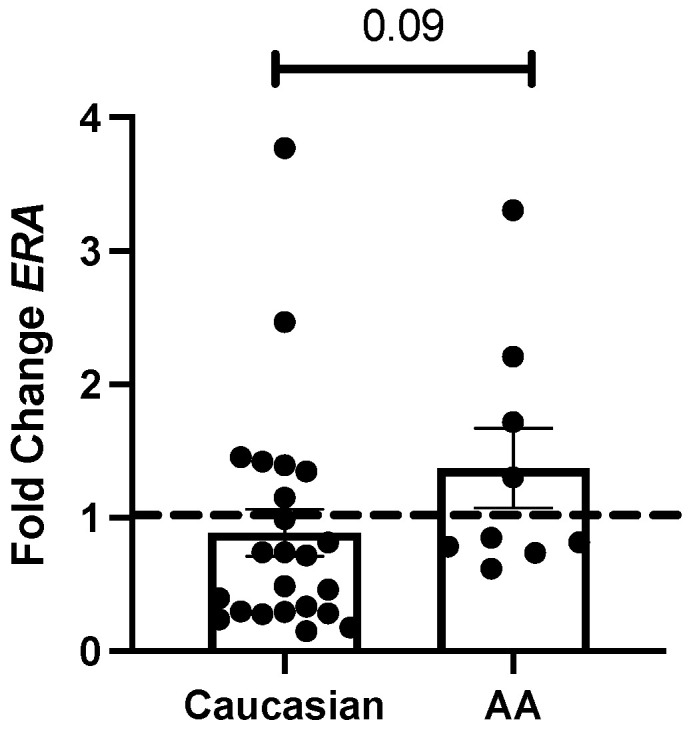
Mean fold change in E2-induced *ERA* steady-state transcript levels at 24 h in AA donors. Dotted line represents E2-treated tissue group normalized to vehicle-treated dermal tissue. Statistical testing was performed using a linear mixed model (LMM) approach, with linear contrasts from the model and additional LMMs that included fixed effects for race, time, and the race-by-time interaction.

**Table 1 biomedicines-12-00182-t001:** Donors’ demographic information.

Donor Characteristics	All	AA	Caucasian	Hispanic	Cambodian
Race/Ethnicity, *n* (%)	34 (100)	9 (26.5)	23 (67.7)	1 (2.9)	1 (2.9)
Age, mean (SD) ^+^	44.8 (10.1)	46.0 (8.05)	44.2 (11.2)	47 *	41 *
Sex, Female, *n* (%)	33 (97.1)	9 (100)	22 (95.7)	1 (100)	1 (100)

* Standard deviation (SD) not included since n = 1. ^+^ One subject missing age which was imputed using the median.

**Table 2 biomedicines-12-00182-t002:** Median vehicle-treated relative quantity transcript levels over time, interquartile range (IQR).

Time (h)	*TGFB1*	*TGFB2*	*Col IA2*	*FN*	*IL-6*	*Col IIIA1*	*ERA*
24	0.09029 (0.05)	0.0006016 (0.0006)	0.002437 (0.004)	0.0008681 (0.0007)	0.01541 (0.02)	0.01746 (0.02)	0.0001013 (0.0001)
48	0.05296 (0.05)	0.0004174 (0.0005)	0.002102 (0.003)	0.001085 (0.001)	0.01116 (0.02)	0.02533 (0.03)	0.00005607 (0.00005)
72	0.08273 (0.08)	0.0004377 (0.0008)	0.002385 (0.004)	0.001063 (0.001)	0.02205 (0.04)	0.01289 (0.01)	0.00006435 (0.0001)

**Table 3 biomedicines-12-00182-t003:** Median E2-induced fold change steady-state transcript levels over time, interquartile range (IQR).

Time (h)	*TGFB1*	*TGFB2*	*Col IA2*	*FN*	*IL-6*	*Col IIIA1*	*ERA*
24	1.098 (0.4)	0.9472 (0.6)	0.7869 (0.4)	0.8938 (0.6)	1.061 (1.2)	0.8973 (0.4)	0.764 (1.0)
48	1.247 (0.7)	1.078 (0.8)	1.269 (0.9)	1.079 (0.7)	1.207 (1.1)	1.081 (0.7)	1.51 (1.4)
72	1.134 (0.7)	0.9632 (0.6)	1.087 (0.8)	0.9078 (0.9)	1.2 (0.9)	1.053 (0.8)	0.8473 (0.9)

**Table 4 biomedicines-12-00182-t004:** Correlations between E2-induced profibrotic mediator expression at 48 h.

	*TGFB1*	*TGFB2*	*IL-6*	*Col IA2*	*Col IIIA1*	*ERA*	Baseline *ERA*
*FN*	**0.55**	0.273	0.280	**0.678**	**0.499**	**0.347**	0.305
*p*	<0.001	0.006	0.004	<0.001	<0.001	<0.001	0.003
adj *p*	**<0.001**	0.168	0.125	**<0.001**	**<0.001**	**0.011**	0.094
*TGFB1*		0.29	0.084	**0.471**	0.175	**0.315**	0.155
*p*		0.003	0.399	<0.001	0.076	0.001	0.144
adj *p*		0.096	1	**<0.001**	1	**0.037**	1
*TGFB2*			0.151	0.278	0.017	0.257	0.015
*p*			0.128	0.004	0.865	0.009	0.889
adj *p*			1	0.105	1	0.252	1.000
*IL-6*				0.205	0.031	0.035	0.151
*p*				0.033	0.752	0.726	0.155
adj *p*				0.929	1	1	1
*Col IA2*					**0.439**	**0.382**	0.127
*p*					<0.001	<0.001	0.201
adj *p*					**<0.001**	**0.003**	1.000
*Col IIIA1*						0.073	0.074
*p*						0.462	0.468
adj *p*						1	1
*ERA*							**−0.457**
*p*							<0001
adj *p*							**<0.001**

Bolded values meet statistical significance.

**Table 5 biomedicines-12-00182-t005:** Correlations between vehicle-treated and E2-treated profibrotic mediator transcript levels.

Genes	Correlation (r)	*p*-Value
*FN*	**0.219**	**0.045**
*TGFB1*	**0.279**	**0.011**
*TGFB2*	0.163	0.135
*IL-6*	**0.250**	**0.021**
*Col IA2*	0.189	0.058
*Col IIIA1*	**0.208**	**0.043**

Bolded values meet statistical significance.

## Data Availability

The data presented in this study are available in the article.

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
