# Peer review of "The Relationship between Time, Race, and Estrogen Receptor Alpha in Estradiol-Induced Dermal Fibrosis"

_biomedicines, 2024, doi:10.3390/biomedicines12010182_

Round 1
Reviewer 1 Report
Comments and Suggestions for Authors
It is a quite interesting study. The topic of this manuscript falls within the scope of Biomedicine Journal This is the first study to investigate the influence of donor characteristics and Estrogen receptor alpha(ERA) steady state transcript levels on E2-induced proinflammatory and profibrotic mediators. On the basis of obtained results, the Authors concluded that ERA, estradiol exposure time, and race/ethnicity contribute to estradiol-induced dermal fibrosis.
The Authors have presented sufficient data. The appropriate table and figures have been provided. The article is easy to read and logically structured. The methods are adequately described. The authors used appropriate statistical methods. The conclusions are consistent with the presented evidence and arguments. Additionally, the Authors added good limitations to their study.
Author Response
Thank you to the reviewer for your comments and support of this manuscript.
Reviewer 2 Report
Comments and Suggestions for Authors
In this work, Wolf and colleagues explored the impact of estradiol on dermal fibrosis through the human skin organ culture model. The role of ethnicity of the donor and estrogen receptor alpha expression has been assessed.
The paper is well organized and the experimental design is easy to follow.
Statistical treatment is valid and conclusions are supported by data.
The following comments could help solve minor issues.
- Define each acronym at first use (i.e. E2 is not defined in the abstract)
- Reformat Table 1
- Revise English language throughout the manuscript
- Reformat the character in the reference list, according to the Journal guidelines
- Add a graphical abstract to catch the reader's attention
Comments on the Quality of English LanguageThe quality of English language should be improved, but it is comprehensible for the readers.
Author Response
Pleae see the attachment.

Reviewer 3 Report
Comments and Suggestions for Authors
The authors try to correlate E2 treatment with time, race and fibrosis. The authors present their data well, but their data does not support their conclusion. They point out that there is a very high variability in their results, and then they try to make conclusions on trends and some borderline statistical differences. When one looks at the data, especially figures 1 and 2, most of the values are clustered at the bottom, with a few outliers that influence their results. Clearly, the data is not in a normal distribution. Do they account for nonparametric statistics? Even if there are significant differences, the data do not seem to have biological significance. Can they reproduce their findings in repeat experiments? This is very important in light of their high variability. I would not accept their conclusions unless the studies were repeated.
Another problem is that they only evaluated mRNA expression. It would be more valuable to have protein data.
Reviewer 4 Report
Comments and Suggestions for Authors
In skin, estradiol promotes profibrotic and proinflammatory cytokines, contributing to extracellular matrix (ECM) deposition. However, the magnitude of response differs. Using the human skin organ culture model, the authors evaluated donor characteristics and correlations that contribute to estradiol-induced interleukin 6 (IL-6), transforming growth factor beta 1 and 2 (TGFB1 and TGFB2), 14 collagen IA2 (Col IA2), collagen IIIA1 (Col IIIA1), and fibronectin (FN) expression. Therefore, the authors suggests that estrogen receptor alpha ERA, estradiol exposure time, and race/ethnicity contribute to estradiol-induced dermal fibrosis.
What is striking about this manuscript is that the authors think that PCR alone is able to answer such important questions. During the writing of the work they show a detailed series of data coming from this technique, making it clear that only this method is able to answer certain questions.
Frankly, I don't understand all this because the authors forget that there are other techniques, primarily morphological ones, which can be added to the work and perhaps in some way satisfy the curiosity of readers who do not come from certain backgrounds and therefore also suffer from a possible interpretation some data. I believe that better organizing the work taking into account these suggestions could satisfy the large audience of readers that characterize this journal.
Comments on the Quality of English LanguageModerate editing of English language required
Reviewer 5 Report
Comments and Suggestions for Authors
My decision is 'Accept after minor revisions':
The authors explore the impact of estradiol on dermal fibrosis through the human skin organ culture model. The authors claim that this is the first study to evaluate the influence of age and ethnicity (self-reported) which is probably true.
The study is well performed and the paper is well-written, but limitations should be mentioned in more detail.
When characteristics are presented and there is only one individual in a group, or only 2, grouping seems unnecessary.
E.g. The results of the 1 male patient may add inaccuracies in the statistics considering the differences in receptors or response to estrogens between males and females. Likewise, only one patient was >70 and one with unknown age. Is it necessary to have a separate group for >70? Perhaps regroup?
And the authors don't present numbers but only % which hides the fact that there were very few representatives in some groups. Furthermore, 1 individual is given as 2.9% in one row and 3% in another. These small inaccuracies may to some readers seem intentional, therefore I suggest more transparency when presenting results and a more thorough description of the limitations, i.e. was the number of participants big enough to obtain robust results?
Round 2
Reviewer 3 Report
Comments and Suggestions for Authors
The authors have addressed my concerns. While there is still a potential for a type 2 statistical error, the paper is worth publishing.
Reviewer 4 Report
Comments and Suggestions for Authors
The authors have answered correctly at all my questions.
Comments on the Quality of English LanguageModerate editing of English language required